# Transcriptional Controls for Early Bolting and Flowering in *Angelica sinensis*

**DOI:** 10.3390/plants10091931

**Published:** 2021-09-16

**Authors:** Mengfei Li, Jie Li, Jianhe Wei, Paul W. Paré

**Affiliations:** 1State Key Laboratory of Aridland Crop Science, Gansu Agricultural University, Lanzhou 730070, China; lijie9654@163.com; 2Institute of Medicinal Plant Development, Chinese Academy of Medical Sciences & Peking Union Medical College, Beijing 100193, China; 3Department of Chemistry and Biochemistry, Texas Tech University, Lubbock, TX 79409, USA; paul.pare@ttu.edu

**Keywords:** *Angelica sinensis*, early bolting and flowering, transcriptomic analysis, gibberellin metabolism, sucrose metabolism

## Abstract

The root of the perennial herb *Angelica sinensis* is a widely used source for traditional Chinese medicines. While the plant thrives in cool-moist regions of western China, early bolting and flowering (EBF) for young plants significantly reduces root quality and yield. Approaches to inhibit EBF by changes in physiology during the vernalization process have been investigated; however, the mechanism for activating EBF is still limited. Here, transcript profiles for bolted and unbolted plants (BP and UBP, respectively) were compared by transcriptomic analysis, expression levels of candidate genes were validated by qRT-PCR, and the accumulations of gibberellins (GA_1_, GA_4_, GA_8_, GA_9_ and GA_20_) were also monitored by HPLC-MS/MS. A total of over 72,000 unigenes were detected with ca. 2600 differentially expressed genes (DEGs) observed in the BP compared with UBP. While various signaling pathways participate in flower induction, it is genes associated with floral development and the sucrose pathway that are observed to be coordinated in EBF plants, coherently up- and down-regulating flowering genes that activate and inhibit flowering, respectively. The signature transcripts pattern for the developmental pathways that drive flowering provides insight into the molecular signals that activate plant EBF.

## 1. Introduction

*Angelica sinensis* (Oliv.) Diels (Family Umbelliferae) is a perennial herb distributed mainly in cool-moist regions of western China at elevations ranging from 2200 to 3000 m [1,2,3]. Roots (Danggui) are prepared as a traditional Chinese tonic reported to nourish the blood and harmonize vital energy [4]. Over 140 root metabolites have been identified, including polysaccharides, organic acids, phthalides, and essential oils [5,6]. These compounds confer pharmacological activities including: anti-inflammatory, antioxidant, anticancer, and cardio-cerebrovascular effects [7,8,9].

Due to an increasing demand for traditional Chinese medicines, *A. sinensis* is farmed to meet commercial demand [3]. For industrialized planting, seeds are sown in early summer, plants are collected in Fall and overwinter indoors; the following spring, seedlings are planted for vegetative growth and are either harvested in Fall of the second year to obtain non-lignified roots or kept in the field till mid-summer of the third year for seed collection (Appendix A) [10]. Early bolting and flowering (EBF) occurs in the second year in up to 40% of the plants, substantially reducing root yield and quality due to lignifications of roots and degradations of bioactive compounds [1,11,12]. For the EBF to occur, the plant must experience vernalization and long-day (LD) conditions; thus, avoiding vernalization or LD conditions can reduce EBF [13,14,15]. In addition, the EBF is also affected by varieties [3], seedling age and weight [16], latitude and longitude [17], and soil conditions [18,19].

The transition from vegetative growth to flowering involves multiple signaling pathways that are transcriptionally regulated including: photoperiodic, autonomous/vernalization, sucrose, and gibberellin (GA) pathways [20]. All pathways converge by increasing the expression of the two meristem identity genes: *SUPPRESSOR OF OVEREXPRESSION OF CONSTANS1* (*SOC1*) that is also known as *AGAMOUSLIKE 20* (*AGL20*) and *LEAFY* (*LFY*). *SOC1* and *LFY*, in turn, regulate the floral homeotic genes to produce the floral organs [20,21]. The photoperiodic pathway is initiated by phytochromes and cryptochromes. The interaction of photoreceptors with a circadian clock activates the expression of the gene *CONSTANS* (*CO*) that encodes a zinc-finger transcription factor that promotes flowering. In the dual autonomous/vernalization pathway, flowering occurs either in response to internal signals, the production of a fixed number of leaves, or to low temperatures that reduces the expression of the flowering repressor gene *FLOWERING LOCUS C* (*FLC*). The sucrose pathway reflects the metabolic state of the plant and sucrose stimulates flowering by increasing *LFY* expression. Lastly, the GA pathway can participate in early flowering and for flowering under noninductive short days.

Besides the previously mentioned developmental pathways that promote flowering, *Arabidopsis* mutants that exhibit early flowering have revealed the involvement of genes that repress flowering. For example, *EMBRYONIC FLOWER1* (*EMF1*) and *EMF2* act as strong flowering repressors [22]; *CURLY LEAF* (*CLF*) by preventing the expression of the floral meristem identity gene *AGAMOUS* (*AG*) during vegetative growth [23]; *EARLY BOLTING IN SHORTDAYS* (*EBS*) participates in the regulation of flowering time by specifically repressing the expression of *FLOWERING LOCUS T* (*FT*) [24]; *Cyclic dof factor 2* (*CDF2*) delays flowering by repressing *CO* transcription [25]; and *MicroProtein 1A* (*MIP1A*) and *MIP1B* repress flowering by forming heterodimeric complexes [26].

Currently, little research focuses on physiological and molecular changes during vernalization of *A. sinensis* seedling, and researches related to growing stage are limited. Specifically, Lu et al. [27] reported that the levels of soluble sugars and protein decreased during the growing stage. Yu et al. [28] found that 5094 genes were differentially expressed in the apical meristem of plants presenting vegetative growth compared to flower buds of early flowering plants, and 13 DEGs were involved in photoperiodic, vernalization, sucrose, and GA pathway. Our previous studies found that 558 genes co-expressed during the four photoperiodic stages of plant growth and 38 DEGs were involved in photoperiodic, hormone signaling, carbohydrate metabolism, and floral development [29].

To date, although the levels of amino acids, GA_3_, zeatin riboside and polyamines, and the activities of peroxidase and polyphenoloxidase in bolting plants (BP) compared to unbolted plants (UBP) have been investigated [27], early bolting-dependent changes that impact genes expression and GAs metabolism have not been investigated and identified. In this study, the BP and UBP were measured by transcriptomic analysis and 40 DEGs associated with EBF were mapped on pathways involved in flowering control. Gene expression levels were validated with qRT-PCR, and down-stream GA metabolites were profiled by HPLC-MS/MS.

## 2. Materials and Methods

### 2.1. Plant Material

Mature seeds of 3-year-old *A. sinensis* (Mingui No. 1) were permitted to collect from the county-owned garden located in Minxian county (2520 m a.s.l.; 34°28′33″ N, 104°05′51″ E) of Gansu province, P. R. China in July 2017. The species was identified by professor Ling Jin (Gansu University of Chinese Medicine, Lanzhou, Gansu, China). A voucher specimen (No. 20200182) was deposited in the herbarium of College of Life Science and Technology, Gansu Agricultural University, Lanzhou, Gansu, China. Seeds were pre-treated in water (30 °C) for 24 h and sown at a soil depth of 0.5 cm located in Minxian county (2730 m a.s.l.; 34°28′8″ N, 104°36′22″ E) in June 2018. Seedlings were dug up in October 2018, aired in the shade for approximately 15 days and then stored in a natural-rain-proof environment for the winter.

On April 3, 2019, the stored seedlings (root tip diameter 4.5–5.0 mm) were transplanted into pots (diameter 17 cm, depth 20 cm; one seedling per pot) with nutrition matrix and seedlings were greenhouse grown with controlling matrix volumetric moisture content of 60–70%, light condition of 10–12 h per day and air temperature 15–22 °C. No additional fertilizer was applied after the transplant. With plant growth and development, some plants began to initiate flower bud differentiation and then BF, other plants kept on vegetative growth with NBP. On 3 July 2019, samples including the second-tip leaves and lateral roots (1:1, g/g fresh weight) from BP and UBP (Appendix A) were collected (*n* = 20 plants with homogeneous growth potential) and then flash frozen in liquid nitrogen for transcriptomic analysis and GA metabolite analysis.

### 2.2. Total RNA Isolation and Illumina Sequencing

Total RNA samples were isolated according to our previous literature [29]. RNA sequencing was conducted using an Illumina HiSeq^TM^ 4000 platform by Gene De novo Biotechnology Co., Ltd. (Guangzhou, China).

### 2.3. Sequence Filtration, Assembly and Unigene Expression Analysis

Sequence filtration, assembly, and unigene expression analysis were conducted according to our previous literature [29]. Briefly, raw reads were filtered by removing reads containing adapters, unknown nucleotides and low-quality bases. De novo assembly of clean reads used a Trinity software [30]. The expression level of each transcript was normalized to reads per kb per million (RPKM) value [31]. In this study, the level of differential expression for each transcript with a criterion of |log_2_ (fold-change)| ≥ 1 and *p* value ≤ 0.05 to identify DEGs between BP and UBP.

### 2.4. Basic Annotation of DEGs and Gene Cluster Analysis

Unigenes were annotated against the databases including: NCBI non-redundant protein (NR), Swiss-Prot protein, Kyoto Encyclopedia of Genes and Genomes (KEGG), euKaryotic orthologous groups of proteins (KOG), and gene ontology (GO) by using a BLASTx procedure with an e-value ≤ 10^−5^ [32]. Molecular Evolutionary Genetics Analysis (MEGA) 7.0 was used for the gene cluster analysis (Appendix A).

### 2.5. qRT-PCR Validation

The relative expression levels (RELs) were validated according to our previous literature [29]. Briefly, total RNA was extracted using a plant RNA kit. Primer sequences of the 40 DEGs (Appendix A) were designed in primer-blast of NCBI. First-strand cDNA was synthesized using a FastKing RT kit. PCR amplification was carried out using a SuperReal PreMix. *Actin* was used as an internal reference and the REL was calculated using a 2^−*ΔΔ**Ct*^ method [33].

### 2.6. GA Quantification and Identification

Samples were ground into powder in liquid nitrogen, 1.0 g sample was soaked in acetonitrile (10 mL) and agitated at 4 °C for 8 h, then centrifuged at 13,000 r/min and 4 °C for 5 min. Following exhaustive extraction (×3), the upper portion was pooled and dried with nitrogen gas in the dark. Dried residue was re-dissolved in methanol (400 µL) and filtered with 0.22 μm durapore membrane. The reference standard of the 5 GAs (GA_1_, GA_4_, GA_8_, GA_9,_ and GA_20_) was dissolved in methanol to make concentrations 0.1, 0.2, 0.5, 2, 5, 20, 50, and 200 ng/mL. Samples (2 µL) were quantified and identified using a HPLC (Agilent1290, USA)-MS/MS (QTRAP 6500, AB SCIEX, USA) by Shanghai Biotree biotech Co., Ltd. (Shanghai, China). Methanol (A) and H_2_O (B) were the gradient elution: 0–1 min 20% A, 1–9 min A increasing up to 80%, 9–10 min 90% A, 10–10.1 min A decreasing down to 20%, and 10.1–15 min 20% A. Representative chromatograms of reference standard of the 5 GAs are shown in Appendix A, and representative chromatograms of the BP and UBP are shown in Appendix A. The content of the 5 GAs was calculated based on calibration curves (Appendix A).

### 2.7. Soluble Sugar Measurement

Soluble sugar was measured using a sulfuric acid-phenol protocol [34]. A dried powder (1.0 g) was soaked in 95% EtOH (25 mL) for 72 h at 22 °C and then centrifuged (4 °C, 8000 r/min, 10 min). Extracts (30 µL) were added into 9% phenol reagent (1 mL), sulfuric acid (3 mL) was added after oscillation and then reacted at 22 °C for 30 min. Absorbance was measured at 485 nm, soluble sugar content was evaluated based on mg of Suc.

### 2.8. Statistical Analysis

All the measurements were performed using three replicates. A t-test for independent samples was performed and SPSS 22.0 was used, with *p* < 0.05 as the basis for significant differences.

## 3. Results

### 3.1. Global Gene Analysis

A robust data set was collected (Appendix A) and after data filtering, 60.7 and 52.4 million high-quality reads were obtained for the BP and UBP, respectively; 44.7 and 37.4 million unique reads as well as 7.8 and 6.4 million multiple reads were mapped. From the 72,502 compiled genes and annotated against the databases including NR, SwissProt, KEGG, KOG, and GO (Table 1 and Table 2, Appendix A), 2645 DEGs were obtained (Appendix A). Of these 2645 DEGs, 369 genes were unidentified by SwissProt, KOG, GO, and KEGG databases. Of the 2276 identified DEGs, 1584 genes with known function were partitioned into being 738 UR and 846 DR. Based on biological function and physiological characteristics, genes were divided into 11 categories: photosynthesis/energy (79), primary metabolism (285), secondary metabolism (80), hormone biosynthesis (34), bio-signaling (201), cell morphogenesis (197), polynucleotide biosynthesis (87), transcription factor (167), translation (119), transport (233), and stress response (102) (Figure 1). Based on flower driving genes characterized in higher plants [20], 40 DEGs (29 UR and 11 DR) were identified as potential regulatory genes for EBF (Figure 1).

### 3.2. DEGs Linked with Bolting and Flowering

Eight DEGs directly participate in floral development including: *SOC1*, *MADS8*, *AGL8*, *AGL12*, *DEFA*, *AP1*, *AP2,* and *ANT* (Table 3). The RELs of these genes were consistent with RPKM values, with up-regulation of 1.1-, 2.4-, 6.8-, 1.1-, 1.3-, and 1.3-fold for *SOC1*, *MADS8*, *AGL8*, *AGL12*, *DEFA,* and *AP1*, respectively, in bolted compared to unbolted plants; down-regulation of 0.6- and 0.9-fold was observed for *AP2* and *ANT* (Figure 2A).

Eleven DEGs associated with sucrose pathway including: Suc metabolism (*SUS1*, *SUS3*, *SUS7*, *INVA*, *INVB*, *INVE,* and *INV Inh*) and starch metabolism (*AMY1.1*, *BAM1*, *BAM3,* and *BAM9*) (Table 3) were transcriptionally regulated so as to favor flowering in BPs. The RELs were consistent with RPKM values, with down-regulated 0.3-fold for the *INV Inh* gene, and up-regulated 1.3- to 6.1-fold for the other 10 genes in the BP compared to the UBP (Figure 2B).

### 3.3. Flower-Regulating DEGs Inarticulately Expressed with EBF

Since GA accumulation can promote flowering, transcripts that encode for GA biosynthesis were screened for up-regulation in EBF plants. The 7 DEGs that are associated with GA signals include: GA biosynthesis (*KO*, *GA2OX1*, *GA2OX6*, *GA2OX8,* and *GA20OX1*) and GA mediated signaling pathway (*GAI* and *GAIP*) (Table 3). The RELs of the 7 genes were consistent with RPKM values, with up-regulated 1.1-, 1.02-, 2.3-, 5.2-, and 1.3-fold for the genes *KO*, *GA20OX1*, *GA2OX6*, *GA2OX8,* and *GAIP*, respectively, in the BP compared to the UBP, and with down-regulated 0.9- and 0.7-fold for the genes *GA2OX1* and *GAI* in the BP (Figure 3A).

The 14 DEGs that are associated with photoperiodic induction include: *CO3*, *COL2*, *COL3*, *COL16*, *FTIP1*, *FD*, *HDR1*, *HD3A*, *MIP1A*, *MIP1B*, *CDF2*, *SVP*, *EFM,* and *AS1* (Table 3). RPKM-based expression values of the 14 genes were validated by qRT-PCR, and their RELs were observed to be consistent with RPKM values, with up-regulated 1.3-, 2.0-, 3.3-, 1.2-, 4.4-, 1.2-, 2.2-, 1.7-, 3.7-, and 1.8-fold for the genes *CO3*, *COL2*, *FTIP1*, *FD*, *HDR1*, *HD3A*, *MIP1A*, *MIP1B*, *CDF2,* and *EFM*, respectively, in the BP compared to the UBP, and with down-regulated 0.7-, 0.98-, 0.9-, and 0.8-fold for the genes *COL3*, *COL16*, *SVP,* and *AS1* in the BP (Figure 3B).

### 3.4. Sucrose and GA Accumulation

Flowering can be initiated by the accumulation of active GAs including GA_1_, GA_3_, GA_4,_ and GA_7_. Interestingly, GA_4_ and GA_1_ as well as the up-stream precursors GA_9_ and GA_20_ had a 3.0-, 1.3-, 5.4-, and 4.2-fold increase in BP while the down-stream inactive forms of GA_4_ and GA_1_, GA_8_ had a 1.5-fold increase in UBP (Figure 4A). Since GA_1_ and GA_4_ exhibit higher floral induction activity than other GAs that are produced in plants [20], an elevated level of GA_1_ and GA_4_ may promote EBF. In contrast, an almost 2-fold decrease in soluble sugars in the BP was unexpected as elevated sugar is usually a driver of flowering [28] (Figure 4B).

## 4. Discussion

The *SUPPRESSOR OF OVEREXPRESSION OF CONSTANS1* (*SOC1*) can integrate signals from the photoperiodism, vernalization, sucrose and GA pathways and regulate the expression of *LFY*, which links floral induction and floral development, when associated with other MADS box genes [35]. MADS box proteins regulate different developmental processes including flowering time, floral meristem identity, and floral organ development [36]. *MADS8,* which is structurally related to the *AGL2* family, is involved in controlling flowering time [37]. *AGL8* promotes early floral meristem identity in synergy with *AP1* and *CAULIFLOWER* [38]. *AGL12* acts as promoter of the flowering transition through up-regulation of *SOC*, *FT,* and *LFY* [39]. *DEFICIENS* (*DEFA*) is involved in the genetic control of floral development [40]. *APETALA1* (*AP1*) and *AP2* are required for the transition of an inflorescence meristem into a floral meristem and promote early floral meristem identity, with *AP1* regulating positively *AG* in cooperation with *LFY*, while *AP2* represses *AG* by recruiting the transcriptional corepressor *TPL* and *HDA19* [41,42]. *AINTEGUMENTA* (*ANT*), a member of the *AP2*-like family, is involved in flower organs initiation and development and mediates *AG* down-regulation [43,44]. Previous studies on *A. sinensis* found that the *SOC1* was down-regulated and the *AG* was up-regulated in 2-year-old plants during transition from vegetative to flower bud differentiation [28]; the *AGL62*, *PMADS1,* and *DEFA* were up-regulated in 3-year-old plants at different growth and development stages [29]. In this study, positive regulators of flowering in the floral development pathway were observed to be up-regulated in EBF plants, while genes that disfavor flowering (*AP2* and *ANT*) were down-regulated, suggesting that transcription regulation of these genes may well be a driver for *A. sinensis* EBF.

Suc and its cleavage products glucose (Glc) and fructose (Fru) are central molecules for cellular biosynthesis and signal transduction throughout a plant’s life cycle [45]. In this study, Suc synthases (SUSs) that are encoded by three *SUS1*, *SUS3,* and *SUS7* genes catalyze a reversible conversion of Suc and UDP to UDP-Glc and Fru [46,47]; Alkaline/neutral invertases (INVs) that are encoded by three *INVA*, *INVB,* and *INVE* genes catalyze an irreversible hydrolysis of Suc to Glc and Fru [48,49,50]; and the invertase inhibitor (*INV Inh*) inhibits the INV activity by forming a complex with INV [51]. Two kinds of amylase enzymes including α-amylase (AMY) and β-amylase (BAM) could respectively produce α-maltose and β-maltose through the hydrolysis of amylopectin and amylose [52]. In this study, four DEGs encoding amylase enzymes include: *AMY1.1,* which can increase enzyme activity via accessory binding sites on the protein surface, *BAM1* and *BAM3,* which play important roles in starch degradation and maltose metabolism, and *BAM9,* which is inactive due to lack the conserved Glu active site [52,53,54]. The *SUS6* and *AMY2* were found to be up-regulated in 3-year-old plants of *A. sinensis* at different development stages [29]. Here, since the genes (*SUS1*, *SUS3*, *SUS7*, *INVA*, *INVB*, *INVE*, *AMY1.1*, *BAM1*, *BAM3,* and *BAM9*) that favor flowering were up-regulated and the *INV Inh* gene that disfavors flowering was down-regulated, transcriptional regulation of sucrose pathway is consistent with EBF.

While genes associated with GA biosynthesis and GA mediated signaling were differentially regulated in BP versus UBP, the genes did not exhibit coherent transcriptional regulation with EBF, suggesting that transcriptional regulation of GA mediated genes is not a driver of early bolting. Previous studies on *A. sinensis* found that the *GA20OX* had no difference change during transition from vegetative to flower bud differentiation [28]; while the *GA2OX1* and *GA2OX**8* were down-regulated at different growth and development stages [29]. For example, with GA mediated signaling, DELLA proteins GA-INSENSITIVE (GAI) and GAIP function as inhibitors by interacting in large multiprotein complexes that repress transcription of GA-inducible genes [55,56,57]. Inconsistent with promoting flowering, the *GAIP* is transcriptionally up-regulated in BP versus UBP. Inconsistency is also observed in genes that encode GA biosynthesis with a subset of genes up-regulated such as *KO*, which catalyzes the conversion of ent-kaurene to kaurenoic acid early in the biosynthetic pathway [58] as well as *GA20OX1,* which converts GA_12_/GA_53_ to GA_9_/GA_20_ [59] later in the pathway (Appendix A), while *GA2OX* catalyzes 2-beta-hydroxylation of GA precursors, rendering them unable to be converted to active GAs is up-regulated under the same condition that promotes flowering (BP). This incoherent transcriptional regulation of GA biosynthesis and signaling for EBF suggests that early bolting may be regulated by events downstream of flowering signaling such as GA and/or sugar accumulation.

While *CONSTANS-LIKE* (*COL*) genes are regulators in the photoperiod pathway and flowering, transcripts in this pathway were also inconsistently induced providing an inarticulate signal for plant flowering, which was in accordance with previous findings with the *CO*, *COL2,* and *COL16* up-regulated while the *COL4* and *COL10* down-regulated in *A. sinensis* [28,29]. For example, while both *CO3* and *COL3* function as floral activators, the two genes were transcriptionally up- and down-regulated, respectively, when comparing BP with UBP. Specifically, *CO3* up-regulates the expression of *Heading date 3a* (*HD3A*) and *FLOWERING LOCUS T-LIKE* (*FTL*) under LD conditions [60,61]. FT-interacting protein 1 (FTIP1) is an essential regulator required for the export of FT protein from the phloem companion cells to sieve elements through the plasmodesmata under LD conditions [62] and was observed to be up-regulated in BP. The FT protein acts as a long-distance signal to induce flowering [63] and *FLOWERING LOCUS D* (*FD*) interacts with FT protein to activate the downstream floral meristem identity genes *AP1* to initiate floral development [64,65]. While this is consistent with flower induction that is observed with BP, there are several transcriptional responses that are not down-regulated as expected. For example, *AS1,* a positive regulator of flowering that binds to the promoter of FT [66], was found to be down-regulated in BP. *CDF2,* a transcriptional repressor that delays flowering by repressing *CO* transcription under LD conditions [25], was found to be up-regulated almost 4-fold in BP compared with UBP. *MIP1A* and *MIP1B*, which repress flowering by forming heterodimeric complexes that sequester CO and COL proteins into non-functional complexes [26], were also found to be up-regulated in BP. Another inconsistent transcriptional response for flowering is up-regulation of *HEADING DATE REPRESSOR 1* (*HDR1*), a flowering suppressor that up-regulates *HD1* in LD conditions [67]. Previous studies on *A. sinensis* also found that the *FTIP1*, *CDF2*, *MIP1A,* and *MIP1B* were up-regulated at different growth and development stages [29]. Again, inconsistent regulation of photoperiod pathway transcripts associated with flowering in BP suggests down-stream signaling involvement in early bolting.

Among the 40 DEGs associated with flowering, 29 genes showed coherent transcriptional regulation with EBF, while 11 genes were incoherent including: *GA2OX6*, *GA2OX8*, *GAIP*, *HDR1*, *COL3*, *COL16*, *AS1*, *CDF2*, *MIP1A*, *MIP1B,* and *EFM*. Extensive experiments have demonstrated that gene expression depends on the plant organ and even on the tissues in each organ [68,69,70]. In this study, the total RNA samples were extracted from the equivalent weight of the leaves and roots from BP and UBP, in theory, the level of gene expression obtained in the experiments is an average value of the expression in the leaves and roots, which could explain the incoherent transcriptional regulation of GA pathway and photoperiodic induction for EBF. For the 11 incoherent genes, their regulatory mechanisms need further validation by detecting gene expression in single organ.

Flowering is a process in which plants transition from vegetative to reproductive growth via a complex pathway of signaling networks. The DEGs observed comparing BP and UBP suggests transcription-based regulation of EBF. Specifically, genes associated with floral development and sucrose signaling are transcriptionally correlated with bolting (Figure 5). For the floral development, *SOC1* can integrate signals from the photoperiodic, GA and sucrose pathways to initiate early floral meristem identity by regulating the over-expression of *LFY*; meanwhile, *AP1* in synergy with *MADS*, *AGL8,* and *AGL12* that are repressed by *AP2* and *ANT*, promote early floral meristem identity. Lastly, the early floral meristem identity induces early bolting and flowering of *A. sinensis* plants. For sugar signaling, over-expression of genes *AMY1.1*, *BAM1,* and *BAM3* enhances starch degradation while differential expression *SUSs*, *INVs,* and *INV Inh* cleavage Suc to Glc and Fru can also promote *SOC1* expression.

## 5. Conclusions

The DEGs observed comparing BP and UBP suggests transcription-based regulation of EBF. This transcriptomic and analysis focuses on four pathways that can mediate a transition from vegetative to reproductive growth: photoperiodic, GA signaling, autonomous, and floral development. While genes associated with EBF have been identified and mapped here, a causative role of these genes in activating and/or regulating EBF will require the knocking out of specific genes via a CRISPR/Cas 9 system.

## Figures and Tables

**Figure 1 plants-10-01931-f001:**
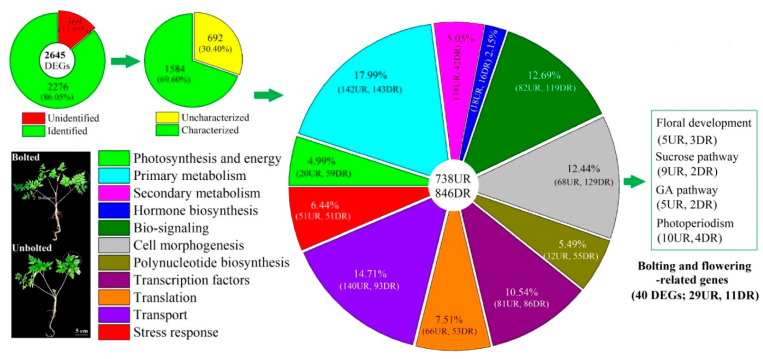
Distribution and classification of differentially expressed genes (DEGs) in bolted versus unbolted *A. sinensis* (UR, up-regulation; DR, down-regulation).

**Figure 2 plants-10-01931-f002:**
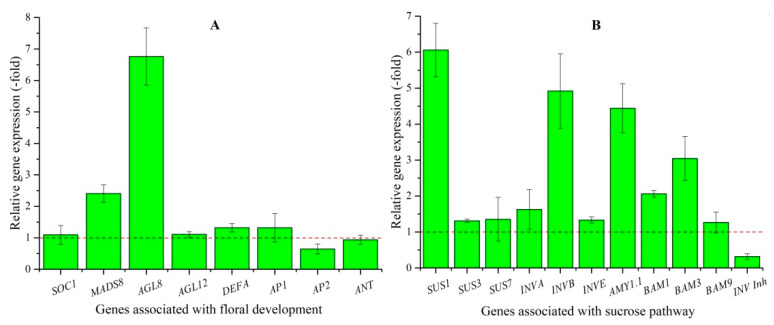
The relative expression level of genes associated with floral development (**A**) and sucrose pathway (**B**) in bolted compared with unbolted plants, as determined by qRT-PCR.

**Figure 3 plants-10-01931-f003:**
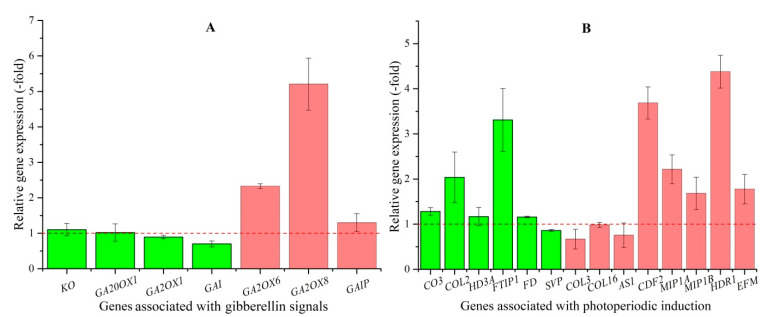
The relative expression level of genes associated with GA (**A**) and photoperiodic pathways (**B**) in bolted compared with unbolted plants, as determined by qRT-PCR. Column highlighted in green represents genes favoring flowering and red represents genes disfavoring flowering.

**Figure 4 plants-10-01931-f004:**
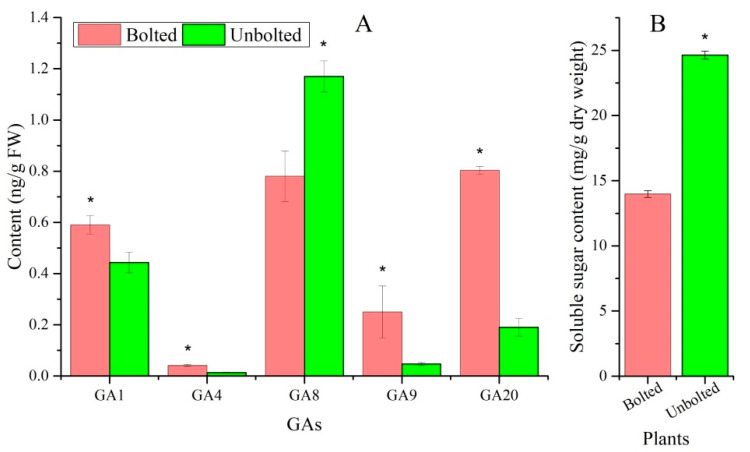
The contents of GAs (**A**) and soluble sugar (**B**) in bolted and unbolted plants, as determined by HPLC-MS/MS. An asterisk (*) represents a significant difference (*p* < 0.05) between BP and UBP.

**Figure 5 plants-10-01931-f005:**
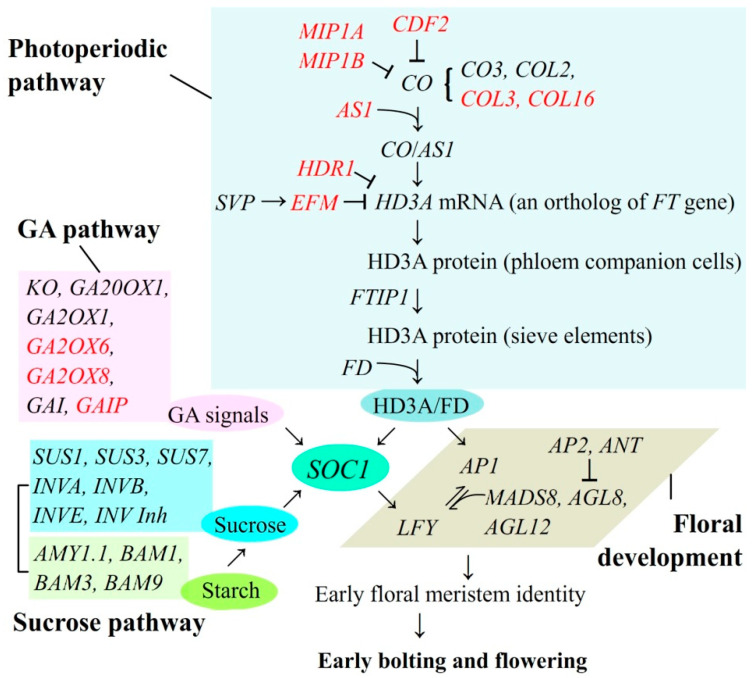
Schematic representation of proposed pathways of DEGs for regulating early bolting and flowering in *A. sinensis*. Genes highlighted in black represent favoring flowering and genes highlighted in red represent disfavoring flowering.

**Table 1 plants-10-01931-t001:** Summary of sequencing data for *Angelica sinensis* transcriptome.

	Bolted	Unbolted
**Unfiltered data**		
Data of reads number (million)	60.73	52.48
Reads length	150	150
GC (%)	44.69	45.12
Data of reads number×read length (million)	9110	7872
Q20 (%)	98.50	98.47
Q30 (%)	95.25	95.18
**Filtered data ^1^**		
Data of reads number (million)	60.66	52.41
Data of reads number×read length (million)	9098	7862
Q20 (%)	98.56	98.53
Q30 (%)	95.34	95.26
**Mapped data ^2^**		
Data of unique mapped reads (million)	44.70	37.40
Data of multiple mapped reads (million)	7.80	6.40
Mapping ratio (%)	86.56	83.57
**Compiled data**		
Total number of unigenes	72,502	
Total Length (bp) (million)	64.14	
N50 (bp)	1534	
Max length (bp)	15,601	
Min length (bp)	201	
Average Length (bp)	884	
GC content (%)	41.17	

^1^ Reads with a quality score < 30 and length < 60 bp were excluded; ^2^ Mapping ratio = (Unique mapped reads + Multiple mapped reads)/Filtered reads.

**Table 2 plants-10-01931-t002:** Database searches for collected *A. sinensis* nucleotide sequences.

BLASTx Searching against Specific Platforms	Values	Percentage (%)
NR	44,708	61.66
SwissProt	30,471	42.03
KOG	22,959	31.67
KEGG	18,056	24.90
GO	12,473	17.20

**Table 3 plants-10-01931-t003:** Bolting/flowering genes differentially expressed in bolted and unbolted *A. sinensis*.

Gene Name	Gene ID	Protein Name	log_2_ Ratio (B_RPKM_/UB_RPKM_)
**Floral development (8)**			
**Genes favoring flowering**			
*SOC1*	XP_017245180.1	MADS-box protein SOC1	1.06
*MADS8*	XP_017257209.1	MADS-box transcription factor 8	7.21
*AGL8*	XP_017244085.1	Agamous-like MADS-box protein AGL8	4.16
*AGL12*	XP_017218759.1	Agamous-like MADS-box protein AGL12	3.42
*DEFA*	XP_017253634.1	Floral homeotic protein DEFICIENS	1.11
*AP1*	AGX01569.1	Floral homeotic protein APETALA 1	4.29
**Genes disfavoring flowering**			
*AP2*	XP_017231882.1	Floral homeotic protein APETALA 2	−6.14
*ANT*	XP_017254585.1	AP2-like ethylene-responsive transcription factor ANT	−3.27
**Sucrose pathway (11)**			
**Genes favoring flowering**			
*SUS1*	XP_017219197.1	Sucrose synthase isoform 1	1.31
*SUS3*	XP_017225961.1	Sucrose synthase 3	1.40
*SUS7*	XP_017244457.1	Sucrose synthase 7	−2.70
*INVA*	CAA76145.1	Alkaline/neutral invertase A, mitochondrial	1.41
*INVB*	XP_017254796.1	Probable alkaline/neutral invertase B	1.22
*INVE*	XP_017258042.1	Alkaline/neutral invertase E, chloroplastic	1.09
*AMY1.1*	XP_017218607.1	Alpha-amylase	1.03
*BAM1*	XP_017219233.1	Beta-amylase 1, chloroplastic	1.62
*BAM3*	XP_017236738.1	Beta-amylase 3, chloroplastic	1.05
*BAM9*	XP_017219710.1	Inactive beta-amylase 9	1.30
**Genes disfavoring flowering**			
*INV Inh*	KZV43516.1	Invertase inhibitor	−1.83
**GA pathway (7)**			
**Genes favoring flowering**			
*KO*	XP_017253618.1	Ent-kaurene oxidase, chloroplastic	2.04
*GA20OX1*	XP_017239190.1	Gibberellin 20 oxidase 1	1.77
**Genes disfavoring flowering**			
*GA2OX1*	API85599.1	Gibberellin 2-beta-dioxygenase 1	−1.41
*GAI*	XP_017238853.1	DELLA protein GAI	−3.49
*GA2OX6*	XP_017243791.1	Gibberellin 2-beta-dioxygenase 6	2.53
*GA2OX8*	XP_017220109.1	Gibberellin 2-beta-dioxygenase 8	1.65
*GAIP*	XP_017217018.1	DELLA protein GAIP	2.15
**Photoperiodic induction (14)**			
**Genes favoring flowering**			
*CO3*	XP_017232180.1	Zinc finger protein CO3	2.58
*COL2*	XP_017231361.1	Zinc finger protein CONSTANS-LIKE 2	3.5
*HD3A*	XP_017216959.1	Protein HEADING DATE 3A	13.41
*FTIP1*	XP_019421416.1	FT-interacting protein 1	2.13
*FD*	XP_017256913.1	Protein FD	3.26
*HDR1*	XP_019170400.1	Protein HEADING DATE REPRESSOR 1	2.12
*SVP*	XP_017245967.1	MADS-box protein SVP	−1.25
**Genes disfavoring flowering**			
*COL3*	XP_017221909.1	Zinc finger protein CONSTANS-LIKE 3	−2.79
*COL16*	XP_017244294.1	Zinc finger protein CONSTANS-LIKE 16	−1.33
*AS1*	XP_017249788.1	Transcription factor AS1	−2.23
*CDF2*	XP_017221059.1	Cyclic dof factor 2	3.03
*MIP1A*	XP_017253198.1	B-box domain protein 30	2.58
*MIP1B*	XP_017253198.1	B-box domain protein 31	3.50
*EFM*	XP_017241902.1	EARLY FLOWERING MYB PROTEIN	1.12

## Data Availability

The datasets generated during the current study are publicly available at National Center for Biotechnology Information (NCBI), with Accession: PRJNA591308 and ID: 591308 (https://www.ncbi.nlm.nih.gov/bioproject/PRJNA591308, accessed on 20 August 2021).

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
