# Peer review of "Transcriptional Controls for Early Bolting and Flowering in Angelica sinensis"

_plants, 2021, doi:10.3390/plants10091931_

Round 1

Reviewer 1 Report

The work submitted by Li and colleagues summarizes the transcriptomic analysis performed to shed light on the molecular basis of early bolting phenotype in A. sinensis, which limits the desired root yield and quality of this perennial herb.

The authors provide sufficient scientific background and the experimental design is solid in its simplicity. The results are carefully presented and discussed. 

I believe that this article meets Plants standards and it would offer interesting food for thought on the subject of the regulation of flowering in non-canonical crops.

Some minor considerations:

- DEGs analysis: can the authors provide a correlation coefficient among RELs obtained from the RNA-seq analysis and the ones of genes validated via qRT-PCR?

- M&M paragraph 2.6: Additional information about HPLC-MS protocol are required to ensure experimental reproducibility;

- Table 3: Include Gene ID codes (for example from NCBI) along with gene names for better understanding;

- LINE 127: "were validated" instead of "was".

Author Response

Thanks for your letter and the news you informed us. Many thanks for the reviewer’s comments that are helpful to improve our paper much better now. We have tried to address and correct each comment. Reviewers’ comments are attached below as well as our responses shown in bold. Revised parts (descriptions) are highlighted in red in the manuscript.

- DEGs analysis: can the authors provide a correlation coefficient among RELs obtained from the RNA-seq analysis and the ones of genes validated via qRT-PCR?

Thanks for your comments on providing the correlation between RPKM values (RELs obtained from the RNA-seq) and genes expression (RELs validated via qRT-PCR) (here, I hope that I can catch you meanings). In fact, the RPKM values (i.e. RELs) of the 40 DEGs obtained from the RNA-seq analysis have been provided in Table 3, meanwhile, the RELs of the 40 DEGs validated via qRT-PCR have been provided in Fig.2 and Fig. 3. As shown in Table 3 as well as Fig. 2 and Fig. 3, RPKM based expression values of the 40 genes (except one gene SUS7) were nearly consistent with RELs validated via qRT-PCR, suggesting there is a positive correlation between RPKM values obtained from the RNA-seq and RELs validated via qRT-PCR.

- M&M paragraph 2.6: Additional information about HPLC-MS protocol are required to ensure experimental reproducibility;

According to your comments, the detail about HPLC-MS protocol has been added in the text: “Samples were grind into powder in liquid nitrogen, 1.0 g sample was soaked in acetonitrile (10 mL) and agitated at 4°C for 8 h, then centrifuged at 13,000 r/min and 4°C for 5 min. Following exhaustive extraction (×3), the upper portion was pooled and dried with nitrogen gas in the dark. Dried residue was re-dissolved in methanol (400 µL) and filtered with 0.22 μm durapore membrane. The reference standard of the 5 GAs (GA1, GA4, GA8, GA9 and GA20) was dissolved in methanol to make concentrations 0.1, 0.2, 0.5, 2, 5, 20, 50 and 200 ng/mL. Samples (2 µL) were quantified and identified using a HPLC (Agilent1290, USA)-MS/MS (QTRAP 6500, AB SCIEX, USA) by Shanghai Biotree biotech Co., Ltd. (Shanghai, China). Methanol (A) and H2O (B) were the gradient elution: 0-1 min 20% A, 1-9 min A increasing up to 80%, 9-10 min 90% A, 10-10.1 min A decreasing down to 20%, and 10.1-15 min 20% A.” (Page 3; lines 138-149)

- Table 3: Include Gene ID codes (for example from NCBI) along with gene names for better understanding;

Thanks for your suggestion, the Gene ID has been added along with gene names in the Table 3. (Page 6; Table 3, lines 202-203).

- LINE 127: "were validated" instead of "was".

The word “were” has instead of “was”. (Page 3; line 131)

Reviewer 2 Report

This contribution focuses on the transcriptional control of early bolting and flowering in Angelica sinensis is a really interesting paper. However, some clarifications and corrections must be made to allow the acceptance of this paper. This remark mainly concerns the parts “introduction” and “plant material”. Their clarification will make it possible to design a more interesting discussion of the results.

First of all, to underline the originality of the proposed work compared to what has already been done for AS, we should write the chapter going from line 71 to line 79. In this chapter, what the authors mean using the term “photoperiodic stage”? Do they mean “growing stage”? Reading this paragraph, it seems that the previous researches have focused on the differences between stages of plant development while this article is mainly interested in differences between plants at the same stage of growth (same age because harvested at the same time) but appearing "normal" or with EBF symptoms. If this is the case, it would be interesting to develop in the discussion the comparison between genes up and down regulated in this study to that highlighted in the previous studies. Line 83, the term “mapped” is inappropriate.

The most astonishing is to note that the samples analyzed are composed of a mixture in equal quantity (by mass) of leaf and root harvested after 3 months of growth in greenhouse. 1/ Mixing leaf and root samples can only make it difficult to compare gene expressions: we know that gene expression is linked to many factors, including those listed by the authors, but it is also dependent on the plant organ and even on the tissues in each organ. The expression obtained in the results presented here is therefore an average value of the expression in the roots and in the leaves. How can this expression give an image of the regulation of the metabolism, when we know that that of the leaves is different from that of the root? And when the expression of certain transcription factors may have an opposite effect depending on the target organ. This has to be mentioned and used to reinforce the discussion. 2/ The harvest is made 3 months after cultivation in a greenhouse. Are all the plants at exactly the same stage of growth and, especially for the 2-year-old bolted plants, have they the same stage of development (same stem elongation ...)?  All these elements must be specified to be sure that the samples prepared are as homogeneous as possible. These details are all the more necessary since only one harvest was made during the differential growth of the plants.

The discussions will only be richer by taking the time to reanalyze the results, taking into account that the quantified expression is an average between that of the leaf and the root. Perhaps the genes which do not present a "coherent transcriptional regulation with EBF" will find their place in the explanation of the regulatory mechanisms.

Author Response

Thanks for your letter and the news you informed us. Many thanks for the reviewer’s comments that are helpful to improve our paper much better now. We have tried to address and correct each comment. Reviewers’ comments are attached below as well as our responses shown in bold. Revised parts (descriptions) are highlighted in red in the manuscript. 

1. First of all, to underline the originality of the proposed work compared to what has already been done for AS, we should write the chapter going from line 71 to line 79. In this chapter, what the authors mean using the term “photoperiodic stage”? Do they mean “growing stage”? Reading this paragraph, it seems that the previous researches have focused on the differences between stages of plant development while this article is mainly interested in differences between plants at the same stage of growth (same age because harvested at the same time) but appearing "normal" or with EBF symptoms. If this is the case, it would be interesting to develop in the discussion the comparison between genes up and down regulated in this study to that highlighted in the previous studies. Line 83, the term “mapped” is inappropriate.

According to your comments, the comparison of DEGs between previous researches and this article has been added in the discussion. The added information is as follow:

1) Previous studies on A. sinensis found that the SOC1 was down-regulated and the AG was up-regulated in 2-year-old plants during transition from vegetative to flower bud differentiation [28]; the AGL62, PMADS1 and DEFA were up-regulated in 3-year-old plants at different growth and development stages [29]. (Page 9; lines 251-255)

2) The SUS6 and AMY2 were found to be up-regulated in 3-year-old plants of A. sinensis at different development stages [29]. (Page 9; lines 271-273)

3) Previous studies on A. sinensis found that the GA20OX had no difference change during transition from vegetative to flower bud differentiation [28]; while the GA2OX1 and GA2OX8 were down-regulated at different growth and development stages [29]. (Page 9; lines 280-283)

4) Which was in accordance with previous findings with the CO, COL2 and COL16 up-regulated while the COL4 and COL10 down-regulated in A. sinensis [28, 29]. (Page 10; lines 298-301)

5) Previous studies on A. sinensis also found that the FTIP1, CDF2, MIP1A and MIP1B were up-regulated at different growth and development stages [29]. (Page 10; lines 319-321)

In addition, the term “mapped” has been revised to “investigated”. (Page 2; line 85).

2. The most astonishing is to note that the samples analyzed are composed of a mixture in equal quantity (by mass) of leaf and root harvested after 3 months of growth in greenhouse. 1/ Mixing leaf and root samples can only make it difficult to compare gene expressions: we know that gene expression is linked to many factors, including those listed by the authors, but it is also dependent on the plant organ and even on the tissues in each organ. The expression obtained in the results presented here is therefore an average value of the expression in the roots and in the leaves. How can this expression give an image of the regulation of the metabolism, when we know that that of the leaves is different from that of the root? And when the expression of certain transcription factors may have an opposite effect depending on the target organ. This has to be mentioned and used to reinforce the discussion. 2/ The harvest is made 3 months after cultivation in a greenhouse. Are all the plants at exactly the same stage of growth and, especially for the 2-year-old bolted plants, have they the same stage of development (same stem elongation ...)? All these elements must be specified to be sure that the samples prepared are as homogeneous as possible. These details are all the more necessary since only one harvest was made during the differential growth of the plants.

Thanks for your constructive suggestions for reinforcing the discussion about the effects of mixture samples on gene expression, as well as the stage of development of bolting and non-bolting plants.

1) According to your comments, the discussion about the effects of mixture samples (different organs) on gene expression has been mentioned and added: “Among of the 40 DEGs associated with flowering, 29 genes were coherent tran-scriptional regulation with EBF, while 11 genes were incoherent including: GA2OX6, GA2OX8, GAIP, HDR1, COL3, COL16, AS1, CDF2, MIP1A, MIP1B and EFM. Extensive experiments have demonstrated that gene expression dependents on the plant organ and even on the tissues in each organ [68-70]. In this study, the total RNA samples were extracted from the equivalent weight of the leaves and roots from BP and UBP, in theory, the level of gene expression obtained in the experiments is an average value of the expression in the leaves and roots, which could explain the incoherent transcriptional regulation of GA pathway and photoperiodic induction for EBF. For the 11 incoherent genes, their regulatory mechanisms need further validation by detecting gene expression in single organ.” (Page 10; lines 324-334)

[68] van der Hoeven, C.; Dietz, A.; Landsmann, J. Variability of organ-specific gene expression in transgenic tobacco plants. Transgenic Res. 1994, 3, 159–166.

[69] Tao, S.Q.; Li, J.; Gu, X.G.; Wang, Y.A.; Xia, Q.; Qin, B.; Zhu, L. Quantitative analysis of ATP sulfurylase and selenocysteine methyltransferase gene expression in different organs of tea plant (Camellia sinensis). Amer. J. Plant Sci. 2012, 3, 51–59.

[70] Che, P.; Lall, S.; Nettleton, D.; Howell, S.H. Gene expression programs during shoot, root, and callus development in Arabidopsis tissue culture. Plant Physiol. 2006, 141, 620–637.

2) In order to ensure the samples prepared are as homogeneous as possible, we selected the plants with homogeneous growth potential at the same growth stages, for example, the BP with the homogeneous stem (bolting) and height, and the NBP with homogeneous height. The information has been added in the Material and method section: “With plant growth and development, some plants began to initiate flower bud differentiation and then BF, other plants kept on vegetative growth with NBP. On July 3, 2019, samples including the second-tip leaves and lateral roots (1:1, g/g fresh weight) from BP and UBP (Figure S2) were collected (n = 20 plants with homogeneous growth potential)”. (Page 3; lines 106-110)   

3. The discussions will only be richer by taking the time to reanalyze the results, taking into account that the quantified expression is an average between that of the leaf and the root. Perhaps the genes which do not present a "coherent transcriptional regulation with EBF" will find their place in the explanation of the regulatory mechanisms.

According to your comments, the discussion on the genes that did not present coherent transcriptional regulation with EBF has been added in the text. Please see the above first response “1.”. (Page 10; lines 324-334)

Round 2

Reviewer 2 Report

The authors have taken into consideration all the suggestions made during the first review and the modifications are well done, except for the paragraph from line 71 to 79. Modifications need to be done so that scientific terms are correctly used. Line 72, "researches on that of..." has to be changed by "and researches on that related to growing stage are limited". Line 74, "photoperiodic stage" has to be changed by "growing stage" (plants have not "photoperiodic stages"). Line 75, change by: " .. in the apical meristem of plants presenting vegetative growth compared to flower buds of early flowering plants and ....".

In the new paragraph from line 318 to line 327, change Line 318:  “29 genes were coherent….” by “29 genes showed coherent…”. Line 321: “that gene expression depends” instead of “dependents”.

When these changes are made, the paper can be accepted for publication.

Author Response

Thanks again for your letter and the reviewer’s comments that are helpful to improve our paper much better now. Reviewers’ comments are attached below as well as our responses shown in bold. Revised parts (descriptions) are highlighted in red in the manuscript.

Reviewers' comments:

Reviewer 2

The authors have taken into consideration all the suggestions made during the first review and the modifications are well done, except for the paragraph from line 71 to 79. Modifications need to be done so that scientific terms are correctly used. Line 72, "researches on that of..." has to be changed by "and researches on that related to growing stage are limited". Line 74, "photoperiodic stage" has to be changed by "growing stage" (plants have not "photoperiodic stages"). Line 75, change by: " .. in the apical meristem of plants presenting vegetative growth compared to flower buds of early flowering plants and ....".

According to your suggestions, the sentence “researches on that of at the photoperiodic stage are limited” has been revised to “and researches on that related to growing stage are limited”; the words “photoperiodic stage” have been revised to “growing stage”; and the sentence “--- the apical meristem of vegetative growth compared to flower buds of early flowering---” has been revised to “in the apical meristem of plants presenting vegetative growth compared to flower buds of early flowering plants---”. (Page 2, lines 74-78)

In the new paragraph from line 318 to line 327, change Line 318:  “29 genes were coherent….” by “29 genes showed coherent…”. Line 321: “that gene expression depends” instead of “dependents”.

The description “29 genes were coherent---” has been revised to “29 genes showed coherent” (), and the word “dependents” has been corrected to “depends” (Page 10, lines 325 and 328).
